# Susceptibility of *Aedes albopictus*, *Ae. aegypti* and human populations to Ross River virus in Kuala Lumpur, Malaysia

**Jolene Yin Ling Fu[1], Chong Long Chua[1], Athirah Shafiqah Abu Bakar[1], Indra Vythilingam[2], Wan Yusoff Wan Sulaiman[2], Luke Alphey[3¤], Yoke Fun Chan[1], I-Ching Sam**  **[1]** \*

**1** Department of Medical Microbiology, Faculty of Medicine, Universiti Malaya, Kuala Lumpur, Malaysia, **2** Department of Parasitology, Faculty of Medicine, Universiti Malaya, Kuala Lumpur, Malaysia, **3** Arthropod Genetics Group, The Pirbright Institute, Woking, United Kingdom

¤ Current address: Department of Biology, University of York, York, United Kingdom
\* jicsam@ummc.edu.my

## Abstract

**Data Availability Statement:** All relevant data are within the manuscript and its Supporting Information files.

### Background

Emerging arboviruses such as chikungunya and Zika viruses have unexpectedly caused widespread outbreaks in tropical and subtropical regions recently. Ross River virus (RRV) is endemic in Australia and has epidemic potential. In Malaysia, *Aedes* mosquitoes are abundant and drive dengue and chikungunya outbreaks. We assessed risk of an RRV outbreak in Kuala Lumpur, Malaysia by determining vector competence of local *Aedes* mosquitoes and local seroprevalence as a proxy of human population susceptibility.

### Methodology/Principal findings

We assessed oral susceptibility of Malaysian *Ae. aegypti* and *Ae. albopictus* by real-time PCR to an Australian RRV strain SW2089. Replication kinetics in midgut, head and saliva were determined at 3 and 10 days post-infection (dpi). With a 3 $\log_{10}$ PFU/ml blood meal, infection rate was higher in *Ae. albopictus* (60%) than *Ae. aegypti* (15%; p<0.05). Despite similar infection rates at 5 and 7 $\log_{10}$ PFU/ml blood meals, *Ae. albopictus* had significantly higher viral loads and required a significantly lower median oral infectious dose (2.7 $\log_{10}$ PFU/ml) than *Ae. aegypti* (4.2 $\log_{10}$ PFU/ml). *Ae. albopictus* showed higher vector competence, with higher viral loads in heads and saliva, and higher transmission rate (RRV present in saliva) of 100% at 10 dpi, than *Ae. aegypti* (41%). *Ae. aegypti* demonstrated greater barriers at either midgut escape or salivary gland infection, and salivary gland escape. We then assessed seropositivity against RRV among 240 Kuala Lumpur inpatients using plaque reduction neutralization, and found a low rate of 0.8%.

### Conclusions/Significance

Both *Ae. aegypti* and *Ae. albopictus* are susceptible to RRV, but *Ae. albopictus* displays greater vector competence. Extensive travel links with Australia, abundant *Aedes* vectors,

**Funding:** This research was funded in whole, or in part, by University Malaya (grant number RK015-2021 to ICS) and the Wellcome Trust (grant number 200171/Z/15/Z to LA). For the purpose of open access, the author has applied a CC BY public copyright licence to any Author Accepted Manuscript version arising from this submission. The funders had no role in study design, data collection and analysis, decision to publish, or preparation of the manuscript.

**Competing interests:** The authors have declared that no competing interests exist.

and low population immunity places Kuala Lumpur, Malaysia at risk of an imported RRV outbreak. Surveillance and increased diagnostic awareness and capacity are imperative to prevent establishment of new arboviruses in Malaysia.

## Author summary

Ross River virus (RRV) is transmitted by mosquitoes and causes outbreaks of fever and joint pain in endemic regions of Australia and some Pacific islands. Arboviruses may spread globally if infected travellers are bitten by potential mosquito vectors in previously unaffected countries with highly susceptible human populations. *Aedes aegypti* and *Ae. albopictus* mosquitoes have not been incriminated as natural RRV vectors, but are highly prevalent in Malaysia and spread other arboviruses such as dengue virus. We showed that Malaysian *Ae. aegypti* and *Ae. albopictus* could potentially transmit RRV from as early as 3 days after an infectious blood meal. *Ae. albopictus* demonstrates higher vector competence for transmitting RRV than *Ae. aegypti*, as it can be infected with lower oral doses, displays higher transmission rates in saliva (100% by 10 days), and higher viral loads in heads and saliva. We tested 240 patient blood samples from Kuala Lumpur, Malaysia and found only a small proportion (0.8%) had neutralizing antibodies against RRV. In summary, Malaysia is at risk of an RRV epidemic due to presence of a naive host population and abundant competent mosquito vectors. Surveillance efforts and epidemic preparedness should include RRV.

## Introduction

Recent re-emerging diseases caused by arboviruses such as chikungunya (CHIKV), dengue (DENV) and Zika (ZIKV) viruses are major public health issues [1]. These arboviruses have adapted to epidemic cycles in which virus transmission is sustained mainly by *Aedes* (*Ae.*) *aegypti* and *Ae. albopictus* mosquitoes. The recent and unexpected global spread of CHIKV and ZIKV from previously geographically restricted areas has increased awareness of future arboviral threats, driven by travel and expansion of *Aedes* distribution [2].

One arbovirus with epidemic potential is Ross River virus (RRV; *Togaviridae* family) which is endemic in Australia and the south Pacific region [3,4]. RRV is within the Semliki Forest serocomplex, which includes CHIKV, O'nyong-nyong virus and Mayaro virus [5]. RRV is usually non-fatal and causes rash, fever, fatigue, polyarthritis and myalgia, but debilitating arthritis may persist for up to 6 months [6]. Large outbreaks of RRV occurred in 1979 and 1980 in the Pacific islands of Fiji, the Cook Islands, Papua New Guinea, New Caledonia, and Vanuatu [7–9]. No major outbreaks have been reported outside of Australia since. The main vectors for RRV in Australia are *Culex annulirostris*, *Ae. vigilax* and *Ae. camptorhynchus* [10]. The main sylvatic hosts are considered to be marsupials such as kangaroos and wallabies, but non-macropod hosts may be important in Fiji and French Polynesia [11,12]. There are currently no approved vaccines or antivirals.

Arboviruses meet various tissue barriers in a mosquito vector before being successfully transmitted to a new host via mosquito bite [13]. Following ingestion in a blood meal, the virus must overcome the midgut infection barrier (MIB) to infect midgut epithelial cells. Then, it must pass through the midgut escape barrier (MEB) to disseminate to other tissue and organs; in particular it must overcome the salivary gland infection barrier (SGIB) to productively infect salivary gland cells. Finally, the virus must pass through the salivary gland escape

barrier (SGEB) to be shed in the saliva, ready to enter the bloodstream of the next host bitten by the mosquito [14,15]. These midgut and salivary gland barriers are critical for determining vector competence, and differ according to the arbovirus-mosquito combination [13].

The risk of local transmission of an arbovirus depends on the simultaneous presence of the virus, competent mosquitoes and susceptible human hosts. As RRV is able to infect multiple mosquito and vertebrate species, it has potential to establish viral transmissions, especially in areas with abundant mosquito vectors and naive populations [16]. While *Ae. albopictus* has been shown to be a competent vector for RRV and may act as a bridge vector for RRV with a potential risk for disease emergence [17], the role of *Ae. aegypti* is less certain [18]. Both *Ae. aegypti* and *Ae. albopictus* are widely distributed in tropical regions, especially in Southeast Asia, and are the main vectors of CHIKV, DENV and ZIKV in Malaysia [19–21]. As there is considerable traffic between Australia and Malaysia, there is a potential that RRV may be introduced by a viraemic traveller. Therefore, the aim of our study was to investigate the potential for an RRV outbreak in Malaysia by determining vector competence of local *Ae. aegypti* and *Ae. albopictus* mosquitoes for RRV. In addition, we assessed RRV serostatus among Malaysians in Kuala Lumpur as a proxy of population susceptibility. By detecting RRV in midguts, heads and saliva of the mosquito vectors, we can also make inferences about the main mosquito tissue barriers faced by RRV.

## Methods

### Ethics statement

The use of human serum samples was approved by the institution's Medical Ethics Committee (no. 2017116–5794). The committee does not require informed consent for retrospective studies of archived and anonymised samples.

### Cell culture and mosquitoes

Vero cells (ECACC 88020401) were maintained in Dulbecco's Modified Eagle media (DMEM) with 10% heat-inactivated foetal bovine serum (FBS; Thermo Fisher Scientific, USA), 2 mM L-glutamine, 1 mM sodium pyruvate, 100 U/ml penicillin and 100 μg/ml streptomycin at 37˚C with 5% $CO_2$.

*Ae. aegypti* and *Ae. albopictus* mosquitoes were collected from the field in Taman Seputeh, Kuala Lumpur. Mosquitoes were maintained under standard insectary conditions of 28 ± 1˚C and 80 ± 10% relative humidity with a 12 h:12 h photoperiod and fed with 10% sucrose supplemented with 0.1% vitamin B. The F3-F4 generation of the mosquitoes was used for virus infection. The mosquitoes were starved for 24 h prior to infection. Mosquito infection work was carried out in an arthropod containment level 2 laboratory.

### Viruses

Ross River virus (RRV) strain SW2089 genotype 3/Eastern (GenBank accession number MN038260), kindly provided by Cheryl Johansen and David Smith from the University of Western Australia, was originally isolated from *Ae. camptorhynchus* mosquitoes in Mandurah, southwest Australia, in 1988. The virus has been passaged eight times in C6/36, Vero cells and BHK cells. The virus stock was stored at -80˚C and used for blood meals.

Chikungunya virus (CHIKV) strain MY/08/065 from the East/Central/South African (ECSA) genotype (GenBank accession number FN295485), was isolated from a patient from Johor, Malaysia in 2008 [19]. CHIKV was passaged three times in Vero cells and was used for the neutralization test.

## Mosquito infection

To determine the optimal infectious dose, *Ae. aegypti* and *Ae. albopictus* mosquitoes were fed blood meals containing RRV at 3 $\log_{10}$, 5 $\log_{10}$ and 7 $\log_{10}$ PFU/ml. RRV was mixed 1:10 with fresh blood donated by one of the co-authors who has no history of RRV infection. The virus-blood mixture was maintained at 37˚C and fed to seven- to eight-day old female *Ae. aegypti* and *Ae. albopictus* in the dark for 1 h using an artificial collagen membrane attached to a Hemotek meal reservoir (Discovery Workshops, UK). After feeding, the mosquitoes were snap frozen for 30 sec and ten engorged mosquitoes were transferred into individual cups and placed in a secure environmental chamber with the previously mentioned conditions. The experiments were conducted according to standard procedures in an arthropod containment level 2 laboratory. We selected 20 mosquitoes at 7 days post-infection (dpi), killed them by freezing, and placed each mosquito in a 1.5 ml zirconium beads tube (Benchmark Scientific, USA) prefilled with 500 μL of serum free DMEM. The mosquitoes were homogenized at 4000 rpm for 15 sec using a microtube homogenizer. The homogenates were stored at -80˚C and used for downstream viral RNA quantification in whole mosquitoes.

## Vector competence analysis

Mosquitoes were orally infected using the optimum infectious doses previously determined and 20 mosquitoes were sampled at 3 and 10 dpi. The mosquitoes were immobilized by snap freezing and had their legs and wings removed to prevent escape. Each mosquito's proboscis was inserted into a pipette tip containing 10 μL of FBS with 10% sucrose and the mosquito allowed to salivate for 1 h. The saliva was then transferred into a PCR tube filled with 40 μL of serum free DMEM. The mosquito's midguts and heads were then dissected and placed individually into a 1.5 ml zirconium beads tube prefilled with 500 μL of serum free DMEM and homogenized. Dissection needles were soaked in 70% alcohol and thoroughly wiped between each mosquito. Vector competence was evaluated by calculating infection rate (number of PCR-positive midguts/number of tested midguts), dissemination rate (number of PCR-positive heads/number of PCR-positive midguts), transmission rate (number of PCR-positive saliva/number of PCR-positive heads), and transmission efficiency (number of PCR-positive saliva/number of mosquitoes fed with infectious blood meal).

## Quantification of RRV viral load

To generate standard curves for RRV quantification, viral RNA was extracted from viral stock using QIAamp Viral RNA Mini Kit (Qiagen, Germany). The cDNA was synthesised with random primers using the Superscript III Reverse Transcriptase System (Thermo Fisher Scientific). PCR was performed using Q5 high fidelity DNA polymerase (NEB, UK), and 10 μM of E1 forward primer (5'-CGATGACGTGGGTACAGAGGA-3') and E1 reverse primer (5'-GTTACCAAGACCAGCACAACCA-3') to generate a 79 base pair fragment [22]. PCR cycling conditions were: 98˚C for 30 sec, then 25 cycles of 98˚C for 10 sec, 69˚C for 30 sec, 72˚C for 30 sec, and 72˚C for 2 min. The PCR fragment was cloned into pJET1.2/blunt cloning vector using CloneJET PCR Cloning Kit (Thermo Fisher Scientific), *in vitro* transcribed using Invitrogen MEGAShortScript Kit, and purified using MEGAClear kit (Thermo Fisher Scientific).

The RRV standard curves were generated using StepOnePlus Real-Time PCR System (Applied Biosystems, USA). The RNA standard was 10-fold serially diluted and added to the 10 μL real-time PCR reaction mixtures containing 4× Taqman Fast Virus 1-Step Master Mix (Thermo Fisher Scientific), 400 nM of E1 forward primer and E1 reverse primer, 250 nM of probe (5'-/56-FAM/TAGAGGGCC/ZEN/AGCCCACCTAACCCACTG/3IABkFQ/-3') [22], and RNAse-free water. PCR cycling conditions were: 1 cycle at 55˚C for 5 min, 95˚C for 20

sec, and 40 cycles at 95˚C for 3 sec and 60˚C for 30 sec. To determine the RRV viral load in the mosquitoes, viral RNA was extracted from mosquito homogenates and 1 μL of RNA was added to the 10 μL real-time PCR reaction mixture as previously mentioned. A blank control and uninfected blood-fed mosquitoes were used as negative controls. Triplicates were performed for each sample. Viral RNA in mosquito samples was quantified by comparing sample cycle threshold (Ct) values to standard curve Ct values. The limit of detection (LoD) of the assay is 1 copy/reaction, equivalent to 2.2 $\log_{10}$ copies/organ and 1.7 $\log_{10}$ copies/saliva. The limit of quantitation (LoQ) of the assay is 100 copies/reaction, equivalent to 4.2 $\log_{10}$ copies/organ and 3.7 $\log_{10}$ copies/saliva.

## Serum samples

We tested seroprevalence of RRV antibodies in residual serum samples collected from inpatients and sent to the diagnostic microbiology laboratory at the University Malaya Medical Centre (UMMC) in Kuala Lumpur. Samples were collected in 2018 and 2019, and we excluded samples sent for dengue (DENV) and CHIKV testing, as both are endemic in Malaysia and have similar clinical presentation to RRV. An older study showed a range of RRV seroprevalence of 0–4% in non-endemic countries in Southeast Asia [23]. Using an estimated prevalence of 3.5% and precision of 2.5%, we calculated a minimum sample size of 208. A total of 240 serum samples were obtained, with 30 samples per age groups <10, 10–19, 20–29, 30–39, 40–49, 50–59, 60–69 and ≥70 years.

## Plaque reduction neutralization test

Serum samples were tested using plaque reduction neutralization test (PRNT) for anti-RRV neutralizing antibodies [24]. Briefly, samples were heat-inactivated at 56˚C for 30 min, then diluted in 1× DPBS from 1:20 to 1:1280 and mixed with 70–80 PFU of RRV diluted in 2% FBS DMEM to a final volume of 200 μL. The virus-antibody mixture was incubated at 37˚C for 1.5 h before inoculation into $10^{5.4}$ Vero cells per well in a 24-well plate. The plate was incubated for 1 h at 37˚C before replacing with plaque medium, 2 parts 3% carboxymethylcellulose and 3 parts 3.5% FBS-DMEM. After 3 days of incubation, cells were fixed with 3.7% of formaldehyde and stained with crystal violet. $PRNT_{50}$ was calculated as the dilution that reduced plaque formation by 50% relative to virus control. As both CHIKV (an alphavirus like RRV) and DENV (flavivirus) are endemic in Malaysia and cause similar symptoms to RRV, we checked for cross-reactivity between RRV, DENV and CHIKV antibodies. We tested 10 DENV-seropositive serum samples, confirmed by Panbio Dengue IgM/IgG ELISA (Abbott, USA), and 6 CHIKV-seropositive serum samples, confirmed by IgG ELISA [25], for ability to neutralize RRV.

Initial screening for neutralizing antibodies against RRV was carried out with samples at 1:10 dilution. Samples that reduced plaques by >75% relative to virus control were retested with 2-fold serial dilutions from 1:20 to 1:320 to determine $PRNT_{50}$. Serum samples with detected anti-RRV antibodies were also tested for specificity against CHIKV by $PRNT_{50}$ and the plates were incubated for 2 days. Virus controls were spiked with serum samples from healthy donors with no serological evidence of past infection of CHIKV and RRV at a dilution of 1:10.

## Statistical analysis

Fisher's exact test (two-tailed) was used to compare infection, dissemination, transmission and transmission efficiency rates for different mosquito species at the same time point or for the same mosquito species at different timepoints. The Mann-Whitney U test was used to

compare distributions or medians of viral loads (with non-normal distributions) between the two mosquito species at each time point. Viral loads with normal distributions were compared with the unpaired t-test. These analyses were performed using GraphPad Prism 9.0 (GraphPad Software, USA). Probit regression was used to calculate the oral infectious doses needed to infect 50%, 75% and 90% of fed mosquitoes ($OID_{50}$, $OID_{75}$ and $OID_{90}$, respectively) [26], using IBM SPSS Statistics v25 (IBM, USA). Differences between OID values of the two mosquito species were considered statistically significant if the 95% confidence intervals (CI) of the OID ratios did not overlap [27]. Using outcomes of dissemination (detectable RRV RNA in heads) and transmission (detectable RRV RNA in saliva), the effects of days post-infection, midgut and head viral loads were determined using multiple regression analysis. Viral loads found to be significant predictors were then evaluated with receiver operating characteristic (ROC) curves for ability to predict dissemination or transmission [28]. Youden's *J* statistic (where *J* = sensitivity + specificity -1) was used to determine the viral load with the highest combined sensitivity and specificity. Two-tailed p-values of <0.05 were considered significant.

Due to the theoretical cross-reaction between alphaviruses, a sample was considered probable RRV-seropositive only if RRV $PRNT_{50}$ titres were $\geq 20$ and additionally CHIKV $PRNT_{50}$ titre was <10 (undetectable). A sample was defined as possible RRV-seropositive with $PRNT_{50}$ titres of $\geq 20$ and detectable anti-CHIKV antibodies, but if CHIKV $PRNT_{50}$ titre was $\geq 4$-fold higher than RRV $PRNT_{50}$, then that sample was considered probable CHIKV-seropositive. Seroprevalence rate was determined by dividing the number of probable cases over the total number of samples tested. Rates are reported with Wilson 95% CI (https://epitools.ausvet.com.au/ciproportion).

## Results

### *Ae. albopictus* is more susceptible than *Ae. aegypti* to oral infection with Ross River virus

*Ae. aegypti* and *Ae. albopictus* were fed infectious blood meals containing 3, 5 and 7 $log_{10}$ PFU/ml of RRV, and infection rates and viral loads in the whole mosquitoes were determined at 7 dpi (S1 Data). Infection rate was significantly higher in *Ae. albopictus* compared to *Ae. aegypti* (Fisher's exact test: 60% vs 15%; p<0.05) at 3 $log_{10}$ PFU/ml, but comparable at 5 $log_{10}$ PFU/ml (75–95%) and 7 $log_{10}$ PFU/ml (100%) (Fig 1A). The $OID_{50}$ and $OID_{75}$ for *Ae. aegypti* were significantly higher than that of *Ae. albopictus*, although the $OID_{90}$ values were similar (Table 1). Increasing the infectious dose from 3 to 7 $log_{10}$ PFU/ml led to increasing viral loads in both mosquitoes, with *Ae. albopictus* demonstrating significantly higher viral loads than *Ae. aegypti* (Fig 1B). Most of the RRV RNA detected in *Ae. aegypti* at infectious doses of 3 to 5 $log_{10}$ PFU/ml fell below the LoQ of the assay. A further observation was the bimodal distribution of viral loads in *Ae. aegypti* at each blood meal titre, with mosquitoes either showing an undetectable/low viral load (at 3 and 5 $log_{10}$ PFU/ml blood meals), or a low/high viral load (at 7 $log_{10}$ PFU/ml). The distributions for *Ae. albopictus* were more tightly clustered. As medians should not be directly compared for binomial distributions, we used the Mann-Whitney U test to compare ranks and distributions of the values, rather than compare medians.

By removing mosquitoes with undetectable RRV RNA, found in the 3 and 5 $log_{10}$ PFU/ml blood meal experiments, the bimodal distributions were no longer present in *Ae. aegypti* and median viral loads could be compared with *Ae. albopictus* (Table 2). The bimodally distributed viral loads in *Ae. aegypti* fed the 7 $log_{10}$ PFU/ml blood meal were also analysed separately as a low (2.8–4.5 $log_{10}$ copies/mosquito) and high (8.7–9.9 $log_{10}$ copies/mosquito) viral load groups. There were no significant differences between the two mosquito species at 3 $log_{10}$ PFU/ml, although only 3 *Ae. aegypti* mosquitoes were infected at this dose. Despite similarly

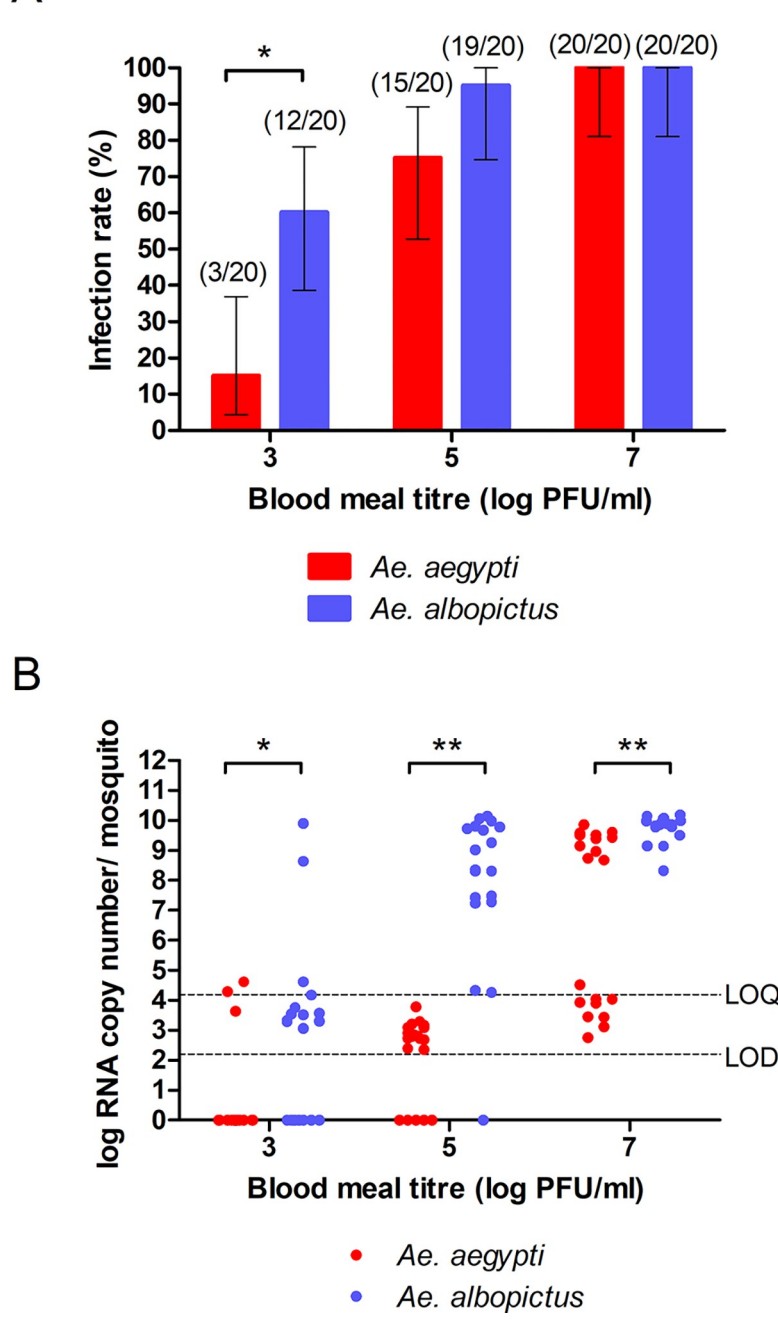

**Fig 1. Infection rates and viral loads of Ross River virus in *Ae. aegypti* and *Ae. albopictus*.** (A) Infection rates of *Aedes* mosquitoes orally infected with three infectious doses (3 $\log_{10}$, 5 $\log_{10}$ and 7 $\log_{10}$ PFU/ml) of RRV at 7 days post-infection. Mosquitoes were considered to be infected with any detectable RNA. Significant differences between the two mosquito species are shown (Fisher's exact test; *, $p<0.05$). Numbers in brackets represent the number of PCR-positive mosquitoes over number of mosquitoes tested. (B) Distributions of viral loads measured by quantitative real-time PCR were compared with the Mann-Whitney U test (*, $p<0.01$, **, $p<0.0001$). LoD, limit of detection, LoQ, limit of quantitation.

**Table 1. Oral infectious doses of Ross River virus for *Ae. aegypti* and *Ae. albopictus*.**

| | Oral infectious dose in $\log_{10}$ PFU/ml (95% CI) | |
| --- | --- | --- |
| | *Ae. aegypti* | *Ae. albopictus* |
| $OID_{50}$* | 4.2 (0.7–5.1) | 2.8 (0–3.6) |
| $OID_{75}$* | 5.0 (2.9–5.8) | 3.6 (1.8–4.3) |
| $OID_{90}$ | 5.7 (4.5–6.8) | 4.3 (3.4–5.2) |

OID, oral infectious dose; PFU, plaque-forming units; CI, confidence intervals.

* $p < 0.05$ difference between the two mosquito species.

high infection rates, *Ae. albopictus* still had significantly higher median viral loads at both 5 and 7 $\log_{10}$ PFU/ml blood meals, even when compared to the *Ae. aegypti* high viral load group.

These findings suggest that *Ae. albopictus* is more susceptible to oral infection with RRV than *Ae. aegypti*. For the optimum oral infectious doses of RRV for further infection, dissemination and transmission experiments, we selected 5 $\log_{10}$ PFU/ml for *Ae. albopictus* and 7 $\log_{10}$ PFU/ml for *Ae. aegypti*, as these doses exceeded the $OID_{90}$ values for each. The range for both these doses is similar to reported levels of viraemia in experimentally infected nonhuman vertebrates [6].

## Malaysian *Ae. aegypti* and *Ae. albopictus* are potential RRV vectors

We determined vector competence (infection, dissemination, transmission rates and transmission efficiency) of *Ae. aegypti* and *Ae. albopictus* mosquitoes (S1 Data) using the previously determined optimum oral infectious doses of 7 $\log_{10}$ PFU/ml and 5 $\log_{10}$ PFU/ml, respectively. Midgut infection rates in *Ae. aegypti* at 3 and 10 dpi ranged between 85–100% (Fig 2A), with similar viral loads (Fig 2D). Virus dissemination increased from 70% at 3 dpi to 100% of the mosquito heads at 10 dpi (Fig 2B), with comparable viral loads (Fig 2E). Transmission was detected in 50% of the saliva at 3 dpi and remained low at 10 dpi at 41% (Fig 2C), with very low viral loads below the LoQ (Fig 2F).

When *Ae. albopictus* was fed with 5 $\log_{10}$ PFU/ml blood meal, all midguts (100%) were infected at 3 dpi (Fig 3A). No significant difference was observed between the midgut viral loads at 3 and 10 dpi (Fig 3D), indicating high susceptibility of *Ae. albopictus* to RRV infection and low MIB. The dissemination rates were high at 95–100% at 3 and 10 dpi, with similar viral loads (Fig 3B and 3E). Transmission rates increased significantly over time from 47% (3 dpi) to 100% (10 dpi) (Fisher's exact test; **, $p < 0.005$), though viral loads remained similar (Fig 3C and 3F). Overall, the results indicate that Malaysian strains of both *Aedes* species are capable of transmitting RRV.

**Table 2. Whole mosquito RRV loads in *Ae. aegypti* and *Ae. albopictus* with detectable virus after different blood meals.**

| Blood meal ($\log_{10}$ PFU/ml) | *Ae. aegypti* | | | *Ae. albopictus* | | | Mann Whitney U | P value |
| --- | --- | --- | --- | --- | --- | --- | --- | --- |
| | n | Median viral load | IQR | n | Median viral load | IQR | | |
| 3 | 3 | 4.3 | 3.6–4.6 | 12 | 3.6 | 3.3–4.5 | 10 | 0.29 |
| 5 | 15 | 2.9 | 2.7–3.2 | 19 | 8.3 | 7.4–9.8 | 0 | <0.0001** |
| 7 | 9 (low) | 3.9 | 3.3–4.0 | 20 | 9.9 | 9.8–10.0 | 0 | <0.0001** |
| | 11 (high) | 9.4 | 9.0–9.6 | | | | 35.5 | 0.001* |

IQR, interquartile range.

Viral loads are measured in $\log_{10}$ RNA copies/mosquito.

*Ae. aegypti* fed a blood meal of 7 $\log_{10}$ PFU/mL showed a bimodal distribution of low and high viral loads (Fig 2), which are shown separately here.

Median viral loads of the two mosquito species are compared at each time point, and significant differences are shown (Mann-Whitney U; *, $p < 0.01$, **, $p < 0.0001$).

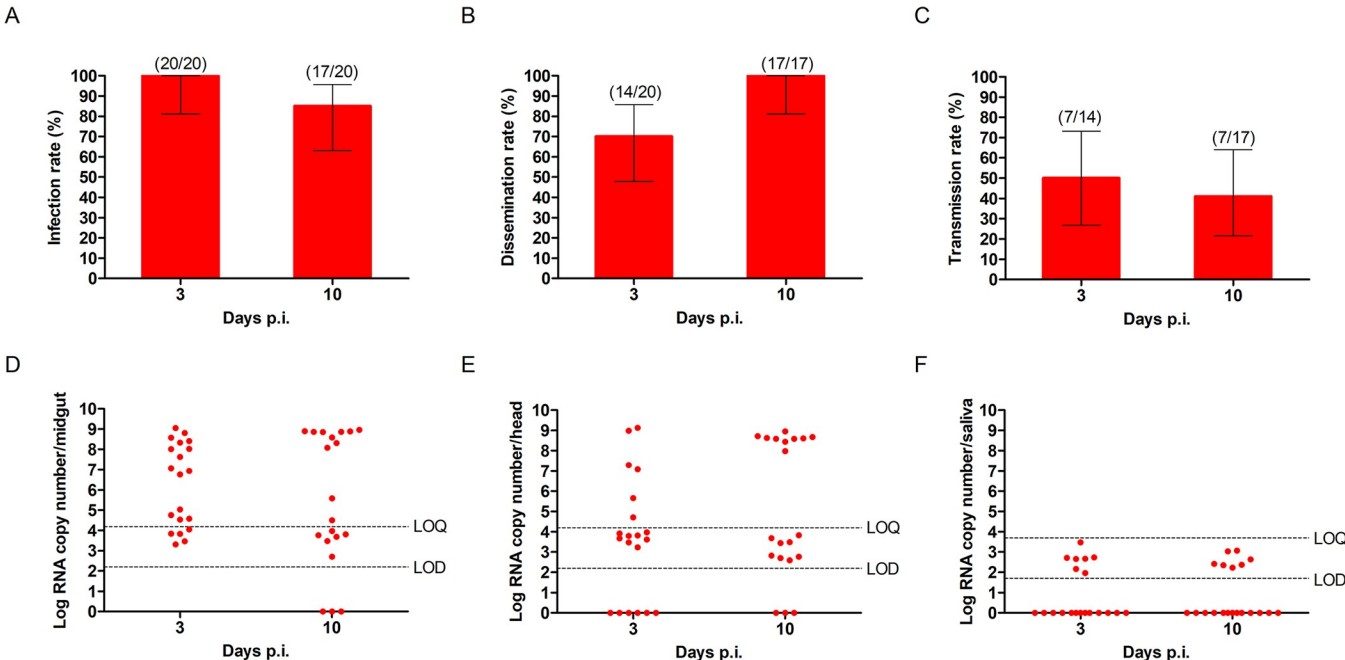

**Fig 2. Vector competence of *Ae. aegypti* for RRV at 3 and 10 days post-infection.** (A) Midgut infection rate, (B) head dissemination rate and (C) saliva transmission rate of *Ae. aegypti* fed with blood meals containing 7 $\log_{10}$ PFU/ml of RRV. Samples were considered to be infected with any detectable RNA. No significant differences were observed (Fisher's exact test). Numbers in brackets represent: infection rate (number of PCR-positive midguts/number of tested midguts), dissemination rate (number of PCR-positive heads/number of PCR-positive midguts), and transmission rate (number of PCR-positive saliva/number of PCR-positive heads). RRV viral loads measured by PCR in (D) midguts, (E) heads and (F) saliva. LoD, limit of detection; LoQ, limit of quantitation.

## *Ae. albopictus* is a more competent vector for RRV than *Ae. aegypti*

We used different RRV blood meal concentrations for the two mosquitoes to ensure infection rates were >90%, which allows comparison of vector competence parameters with a similar starting infection rate (S1 Fig). Both *Ae. aegypti* and *Ae. albopictus* demonstrated high midgut infection rates (S1A Fig), with similar viral loads (S1D Fig). This suggests no significant MIB in both *Aedes* vectors. Dissemination to heads was also similar in *Ae. aegypti* and *Ae. albopictus* at 3 dpi (70% vs 95%) and 10 dpi (100% vs 100%) (S1B Fig), but there were significantly higher viral loads in *Ae. albopictus* heads (S1E Fig). We did not specifically dissect out and include salivary glands with the head, so some of the detectable RRV in heads would be from non-salivary tissue. Nevertheless, this finding suggests either a greater MEB and/or head tissue barrier (which may include the SGIB) in *Ae. aegypti*. Subsequently, for *Ae. aegypti*, only 41–50% of the saliva were positive for RRV at 3 and 10 dpi (S1C Fig), with very low viral loads below the LoQ (S1F Fig). This suggests *Ae. aegypti* presents a greater SGEB to RRV than *Ae. albopictus*, which showed 100% (p<0.005) dissemination at 10 dpi.

As seen in the oral infectious dose experiments, the viral loads in some *Ae. aegypti* organs had a bimodal distribution. This was less evident for *Ae. albopictus*, apart from head viral load at 10 dpi. As transmission is a key endpoint, we removed saliva samples with undetectable RNA in order to compare the mean (normally distributed) viral loads in only the positive samples (S1 Table). When RRV was detectable in saliva, *Ae. albopictus* had significantly higher mean viral loads than *Ae. aegypti* at 3 dpi (3.9 vs 2.6 $\log_{10}$ copies/sample) and 10 dpi (4.7 vs 2.6 $\log_{10}$ copies/sample).

Overall, despite being infected with a viral dose which was 100-fold lower than *Ae. aegypti*, *Ae. albopictus* showed greater competence for RRV, with higher transmission rates and efficiency and viral loads in the saliva.

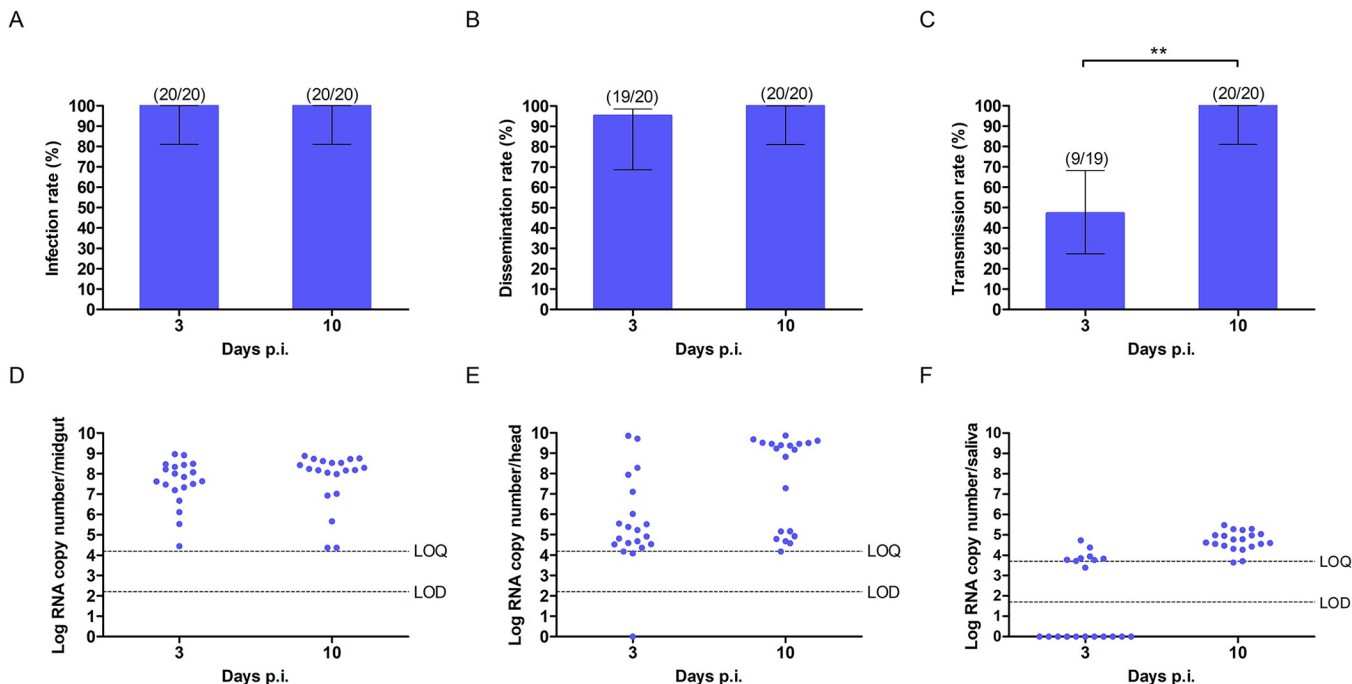

**Fig 3. Vector competence of *Ae. albopictus* for RRV at 3 and 10 days post-infection.** (A) Midgut infection rate, (B) head dissemination rate and (C) saliva transmission rate of *Ae. albopictus* fed with blood meals containing 5 log$_{10}$ PFU/ml of RRV. Samples were considered to be infected with any detectable RNA. Significant differences are shown (Fisher's exact test; **, p<0.0005). Numbers in brackets represent: infection rate (number of PCR-positive midguts/number of tested midguts), dissemination rate (number of PCR-positive heads/number of PCR-positive midguts), and transmission rate (number of PCR-positive saliva/number of PCR-positive heads). RRV viral loads measured by PCR in (D) midguts, (E) heads and (F) saliva. LoD, limit of detection; LoQ, limit of quantitation.

## Viral loads as predictors for dissemination and transmission

There were no instances of positive heads (dissemination) with undetectable midgut viral load, and no instances of positive saliva (transmission) with undetectable virus in midguts or heads. This is consistent with infection of midguts and heads being required for RRV to be potentially transmissible in saliva. We carried out logistic regression to determine the predictive value of

**Table 3. Effects of midgut and head viral loads on dissemination and transmission rates of RRV.**

| Vector | Outcome | Viral load as variable | aOR (95% CI) | P value | AUC (95% CI) | P value | Cut-off value (log$_{10}$ RNA copies/sample) | Sensitivity, specificity |
|---|---|---|---|---|---|---|---|---|
| *Ae. aegypti* | Dissemination | Midgut | 2.2 (1.2–3.8) | 0.009* | 0.84 (0.71–0.96) | 0.002** | 5.3 | 67.7%, 100% |
| | Transmission | Midgut | 1.5 (1.1–2.0) | 0.02* | 0.69 (0.53–0.86) | 0.047* | 5.3 | 85.7%, 65.4% |
| | | Head | 1.2 (1.0–1.5) | 0.09 | - | - | - | - |
| *Ae. albopictus* | Dissemination | Midgut | 0.4 (0.01–14.2) | 0.59 | - | - | - | - |
| | Transmission | Midgut | 1.1 (0.6–1.9) | 0.72 | - | - | - | - |
| | | Head | 1.8 (1.1–2.9) | 0.02* | 0.77 (0.63–0.92) | 0.01* | 6.6 | 58.6%, 90.9% |

aOR, adjusted odds ratio; CI, confidence intervals; AUC, area under the curve; dissemination, detectable RRV RNA in heads; transmission, detectable RRV RNA in saliva. * p<0.05

** p<0.005

midgut and head viral loads on dissemination and transmission rates (Table 3). In *Ae. aegypti*, midgut viral load was a good predictor (AUC of 0.84) of disseminated infection, but a poor predictor (AUC of 0.69) of transmission. In *Ae. albopictus*, viral load in the head was a fair predictor of detectable RRV in saliva (AUC of 0.77). The cut-off values giving the best combined sensitivity and specificity rates were 5.3 $\log_{10}$ RNA copies/midgut for *Ae. aegypti* and 6.6 $\log_{10}$ RNA copies/head for *Ae. albopictus*, but sensitivity rates were low, ranging from 58.6–85.7%.

## Population susceptibility to RRV in Kuala Lumpur

We assessed seroprevalence of residual diagnostic serum from Kuala Lumpur hospital patients as a proxy of population susceptibility. No cross-neutralizing activity against RRV was demonstrated with 6 CHIKV-seropositive and 10 DENV-seropositive serum samples. Of the 240 patient samples tested, 4 were possible RRV-seropositive with $PRNT_{50}$ titres $\geq$20 (range, 20–80). The two samples with RRV $PRNT_{50}$ of 20 had CHIKV $PRNT_{50}$ titres which were 4-8-fold higher, and were considered probable CHIKV-seropositive. The other two samples (RRV $PRNT_{50}$ of 40 and 80) were considered probable RRV-seropositive as CHIKV $PRNT_{50}$ titres were <10. Overall, there was very low probable RRV-seropositivity (2/240, 0.8% [0.2%-3.0%]) in this sample of hospital inpatients from Kuala Lumpur.

## Discussion

There is extensive travel from Australia to Malaysia (1.1 million airline passengers in the year ending June 2019) [29], and an abundance of *Aedes* vectors in Malaysia [30]. We therefore studied vector competence and seroprevalence data as a proxy for population susceptibility to assess risk for RRV transmission in Malaysia.

RRV is thought endemic only in Australia and Papua New Guinea, maintained in nature by native macropods [10]. However, a wide range of mammals and birds have been infected experimentally, and RRV seropositivity has been found in common animals such as cows, goats and horses [31]. Significant RRV outbreaks and enzootic transmission have occurred in regions without marsupials, such as the Pacific Islands and metropolitan Australia [32,33]. High viraemia in humans and low seroprevalence in nonhuman vertebrates during these outbreaks strongly suggest human-mosquito-human transmission [31]. Thus, epidemic RRV transmission is possible without preferred animal hosts, if competent mosquito vectors are abundant, climatic conditions are ideal, and human herd immunity is low [34,35].

We showed that Malaysian field *Ae. aegypti* and *Ae. albopictus* are susceptible to RRV in a dose-dependent manner. Both species potentially transmit RRV, with virus RNA detectable in saliva from 3 dpi. Dose-dependent oral infection is important, as higher dose blood meals increase transmission of CHIKV by *Ae. albopictus* [36]. The $OID_{90}$ of *Ae. aegypti* (5.7 $\log_{10}$ PFU/ml) and *Ae. albopictus* (4.3 $\log_{10}$ PFU/ml) were comfortably below reported viraemia rates of up to 7.6 $\log_{10}$ PFU/ml in experimentally infected mammals [6] and 8 $\log_{10}$ PFU/ml in naturally infected horses [37]. Thus, *Aedes* mosquitoes may facilitate human-mosquito-human transmission, as reported in the Cook Islands RRV outbreak involving *Ae. polynesiensis* [8].

The short extrinsic incubation period (EIP) of 3 days observed in both *Aedes* species here is comparable with the established RRV vector *Ae. vigilax* in Australia [38]. A similarly short EIP in *Ae. albopictus* facilitated the large CHIKV outbreak in Malaysia in 2008 [19]. *Ae. albopictus* is a potential RRV vector in Australia, Asia and the USA [17,39,40], and has comparable vector competence with *Ae. vigilax* [41]. The mean transmission efficiency for enzootic RRV by *Ae. albopictus* may exceed epidemic viruses such as DENV, yellow fever virus and ZIKV; with its opportunistic feeding behaviour and wide host-feeding range, *Ae. albopictus* may serve as a bridge vector to transmit emerging viruses between animals and humans [42].

We further found that *Ae. albopictus* is a more competent RRV vector than *Ae. aegypti*, with lower threshold for oral infection and higher transmission efficiency (100% *vs* 35% at 10 dpi) despite a blood meal containing 100× lower RRV load. Although little is known about RRV and tissue barriers in *Aedes* vectors, there are several studies of other arboviruses [13,43]. We found no significant MIB to RRV in both *Aedes* species. However, *Ae. aegypti* showed either a greater MEB or SGIB than *Ae. albopictus*, resulting in significantly lower virus titres in heads. Our data indicates that a threshold of viral infection is important for midgut escape. A blood meal triggers transient disruption of midgut basal lamina, and an arbovirus must replicate to achieve that threshold quickly enough to pass through and disseminate [15]. Mosquito innate immune responses like the RNA interference system [44] also modulate viral escape from the midgut, as shown for Sindbis virus [45].

The MIB and SGIB require the presence of appropriate receptors for virus binding and entry, which vary within organs. For example, SINV was detected in lateral but not medial salivary lobes of *Ae. aegypti* [46]. This may affect the SGEB as virus is released into apical cavities where saliva is stored before entering salivary ducts. In our study, a SGEB to RRV was evident early (3 dpi) in both *Aedes* species, which by 10 dpi had been overcome in *Ae. albopictus*, but not *Ae. aegypti*. SGEB is influenced by genetic differences within *Ae. aegypti* populations and viral lineages, as shown for CHIKV [47,48]. Two Western equine encephalitis virus strains showed different saliva shedding in *Culex tarsalis*, which was partially overcome by a chimeric recombinant containing structural genes from the more transmissible strain [49]. Thus, *Aedes* species in Malaysia will likely demonstrate varying competence for RRV strains other than SW2089 used in this study.

We found bimodal distributions of viral loads—particularly in *Ae. aegypti*—which may represent distinct mosquito or virus subpopulations. The association between higher *Ae. aegypti* midgut RRV loads with dissemination suggests that a viral load threshold is required to cross the barrier. The basis of intraspecies variation in vector competence is unknown, but involves complex interplay of numerous genes with environmental factors [50]. Virus quasispecies with greater vector adaptation may arise during infection; key envelope protein E1 mutations in CHIKV emerged in *Ae. albopictus* mosquitoes that could transmit CHIKV, but were absent in those that could not [51]. We could further explore this by whole genome sequencing of RRV from successfully infected *Aedes* mosquitoes demonstrating large differences in transmission.

We then assessed RRV seroprevalence as a proxy for population immunity in the event of imported RRV. Anti-RRV antibodies likely persist and protect for life after infection [52]. Our low rate of anti-RRV neutralizing antibodies of 0.8% (0.2%-3.0%) in Kuala Lumpur inpatients indicates susceptibility and low likelihood of unsuspected endemicity. This supports a 1975 study reporting zero seroprevalence in Southeast Asian populations, including Malaysia [23]. In contrast, seropositivity rates of >30% are reported from RRV-endemic Papua New Guinea [23] and Pacific islands where endemicity is suspected [3]. Furthermore, the lack of cross-neutralization by anti-CHIKV antibodies against RRV indicates that CHIKV endemicity in Malaysia will not protect human populations against RRV [53].

There are limitations to our work. Laboratory demonstration of vector competence does not replicate natural conditions, and surveillance of field mosquitoes should be done. To date, *Ae. albopictus* and *Ae. aegypti* have not been incriminated as natural RRV vectors [54], so it would be important to test vector competence of other *Aedes* and *Culex* species found in Malaysia. Single strains of RRV and each mosquito species were tested, and clearly more should be evaluated. There are at least 4 RRV genotypes (G1-G4) with G4 the most recently predominant, although it is most closely related to G3 [55], to which the study RRV strain SW2089 belongs. *Ae. aegypti* from Fiji, Townsville and Jakarta showed good vector competence to RRV [40,54], thus vector competence differences exist among mosquitoes of the same

species from different regions. In addition, since PCR cannot distinguish between live and non-viable virus, future work should determine infectious virus titres to confirm dissemination and transmission; nevertheless, the presence of high viral loads of 4–5 $\log_{10}$ copies in saliva (higher in organs) in *Ae. albopictus* strongly suggests viral replication occurred and that transmission is possible. Our seroprevalence findings are limited by bias from studying a single centre. However, convenience sampling of residual hospital serum gave similar estimates of influenza seroprevalence to cohort studies [56]. Future sampling should be more representative and include rural areas where *Ae. albopictus* predominates [57,58].

In summary, RRV is an arbovirus with epidemic potential to spread beyond its current geographical distribution. We show that Kuala Lumpur fulfils two key factors determining risk for RRV outbreaks: a naive host population and abundant competent mosquito vectors. Virus introduction into suitable conditions by a viraemic traveller, reported in neighbouring Singapore [59], could potentially trigger an outbreak, as seen in the South Pacific in 1979–1980 [8]. The greater vector competence of *Ae. albopictus* over *Ae. aegypti* has implications for disease distribution; the preference of epidemic CHIKV for *Ae. albopictus* led to disproportionate disease incidence in rural areas in Sarawak, Malaysia [60]. The possible roles of animals as reservoirs should be explored, although enzootic transmission is unlikely to be important if these two *Aedes* mosquito species are involved as vectors, as they predominantly feed on humans [61]. Nevertheless, early detection in countries at risk, particularly with travellers from RRV-endemic areas, requires surveillance for RRV, increased diagnostic awareness and laboratory preparedness.

## Supporting information

**S1 Data. Viral load data from oral infection and replication kinetics experiments.**
(XLSX)

**S1 STROBE Checklist.**
(DOCX)

**S1 Fig. Vector competence of *Ae. aegypti and Ae. albopictus* for RRV at 3 and 10 days post-infection.**
(TIF)

**S1 Table. Salivary RRV loads in *Ae. aegypti* and *Ae. albopictus* mosquitoes with detectable virus.**
(DOCX)

## Acknowledgments

We thank Cheryl Johansen and David Smith of the University of Western Australia for providing the Ross River virus isolate.

## Author Contributions

**Conceptualization:** I-Ching Sam.

**Data curation:** Jolene Yin Ling Fu, Athirah Shafiqah Abu Bakar.

**Formal analysis:** Jolene Yin Ling Fu, Chong Long Chua, I-Ching Sam.

**Funding acquisition:** Luke Alphey, I-Ching Sam.

**Investigation:** Jolene Yin Ling Fu, Chong Long Chua, Athirah Shafiqah Abu Bakar, Indra Vythilingam.

**Methodology:** Jolene Yin Ling Fu, Chong Long Chua, Athirah Shafiqah Abu Bakar, Indra Vythilingam, Wan Yusoff Wan Sulaiman, Yoke Fun Chan, I-Ching Sam.

**Project administration:** Jolene Yin Ling Fu, Yoke Fun Chan, I-Ching Sam.

**Resources:** Indra Vythilingam, Wan Yusoff Wan Sulaiman, Yoke Fun Chan, I-Ching Sam.

**Supervision:** Yoke Fun Chan, I-Ching Sam.

**Visualization:** Jolene Yin Ling Fu.

**Writing – original draft:** Jolene Yin Ling Fu.

**Writing – review & editing:** Jolene Yin Ling Fu, Chong Long Chua, Athirah Shafiqah Abu Bakar, Indra Vythilingam, Wan Yusoff Wan Sulaiman, Luke Alphey, Yoke Fun Chan, I-Ching Sam.

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
