## [Decision Letter · Decision Letter 0]

11 Jan 2023

Dear Dr. Sam,

Thank you very much for submitting your manuscript "High susceptibility of Aedes albopictus, Ae. aegypti and human populations to Ross River virus in Kuala Lumpur, Malaysia" for consideration at PLOS Neglected Tropical Diseases. As with all papers reviewed by the journal, your manuscript was reviewed by members of the editorial board and by several independent reviewers. In light of the reviews (below this email), we would like to invite the resubmission of a significantly-revised version that takes into account the reviewers' comments. 

This study is interesting however, some part of the manuscript, especially results and discussion parts need attention. 

We cannot make any decision about publication until we have seen the revised manuscript and your response to the reviewers' comments. Your revised manuscript is also likely to be sent to reviewers for further evaluation.

Sincerely,

Olaf Horstick, FFPH(UK)

Academic Editor

Elvina Viennet

Section Editor

Reviewer's Responses to Questions

**Key Review Criteria Required for Acceptance?**

**Methods**

-Are the objectives of the study clearly articulated with a clear testable hypothesis stated?

-Is the study design appropriate to address the stated objectives?

-Is the population clearly described and appropriate for the hypothesis being tested?

-Is the sample size sufficient to ensure adequate power to address the hypothesis being tested?

-Were correct statistical analysis used to support conclusions?

-Are there concerns about ethical or regulatory requirements being met?

Reviewer #1: The study has met the acceptance criteria listed in Methods.

Reviewer #2: The technics used in this manuscript are adapted but some information are missing.

Authors didn't describe as they determined the number of sera used for the seroprevalence study for the cross-reactivity part.

The analysis by RT-PCR limited the interpretation due to the detection of both dead and infectious viral particles.

Reviewer #3: The aims of the study are clearly stated, and the methods followed are in accordance with the study objectives. They used field populations of the mosquitoes that they tested and an appropriate population size. I would prefer to have a proven vector of the virus as a positive control. I also think that it would be more logical to use the same dose of the virus for the two mosquito species that were tested for the competence or use the two different doses and check the infection-dissemination-transmission-transmission efficiency for both Ae albopictus and Ae aegypti. My only concern is that they didn't use an infectious assay at least for determining the presence of the infectious virus in the saliva.

**Summary and General Comments**

Reviewer #1: The work is nicely done and presented. I am not an English language/grammar expert to make comments on its writing, but scientifically the paper is sound. 

Just few minor comments: 

1. Recommend revising the title. The word "High" seems vague, may be delete this from the title and leave rest of the sentence as is. 

2. Abstract/L28: Suggest changing "Worldwide" to "widely distributed in countries of tropical and subtropical regions" or sth similar. The distribution of chikungunya and Zika is not yet worldwide. 

3. L168-169: "Dissection needles were soaked in 70% alcohol between each mosquito". 70% alcohol disinfect the virus but not really destroy their nucleic acids. When was the control uninfected mosquito dissected - if control mosquito was dissected at first and test mosquitoes later on just soaking the needle in alcohol in-between of dissection, then there might be some possible contamination issues with the procedures. Soaking dissection needle in 10% bleach, then to alcohol is recommended.

Reviewer #2: The manuscript is relatively well written, and the results are interesting. However, some part of the manuscript must be rewritten especially the results and the discussion parts as described in the specific comment. Firstly, I would like to suggest being careful with the results because authors analyzed the body, head and saliva by RT-PCR not by virus detection with cells. Secondly, I really think the second part of the vector competence study must be rewritten. Indeed, authors decided to use different virus titer in the blood meal according with the previous results obtained in the manuscript. That is interesting but that also limited the comparison between both vectors. I recommend to describe this part one vector and after the other one but not to compare both in all this part. Finally, the discussion must be summarized and clarified. I think this manuscript needs major revision before being accepted for publication.

Reviewer #3: Overall, I think it is a good draft. I would suggest discussing more the fact that they used only PCR for determining transmission instead of an infectious assay. In addition, I would add something about the role that the animals play (or not) in spreading the virus and may suggest in the Conclusion if further research is needed regarding the role of animals. Also, in the Conclusion part it should be addressed the fact that the sero-prevalence of the virus in human population is still low, pointing that, for now, the risk is low in Malaysia.

PLOS authors have the option to publish the peer review history of their article (what does this mean?). If published, this will include your full peer review and any attached files.

Reviewer #1: No

Reviewer #2: No

Reviewer #3: No

**Results**

-Does the analysis presented match the analysis plan?

-Are the results clearly and completely presented?

-Are the figures (Tables, Images) of sufficient quality for clarity?

Reviewer #2: I think this part must be rewritten.

Authors should be careful with the interpretation of the results due to the protocoles used.

**Conclusions**

-Are the conclusions supported by the data presented?

-Are the limitations of analysis clearly described?

-Do the authors discuss how these data can be helpful to advance our understanding of the topic under study?

-Is public health relevance addressed?

Reviewer #2: The discussion must be clarififed and summarized.

Reviewer #3: They don't have a conclusion part, but a summary as a last paragraph in Discussion. I think that they did well to mention the probable risk that the virus may pose, but the fact that they did find a very low sero-prevalence of the virus in the human population in Kuala Lumpur should be further address and maybe suggest further research regarding the role of animals in the circulation of the virus in the area.

**Editorial and Data Presentation Modifications?**

Reviewer #2: Major modifications needed

Reviewer #3: I would suggest a minor modification to the title of the article "High susceptibility of Aedes albopictus and Ae. aegypti to Ross River virus and sero-prevalence of human population in Kuala Lumpur, Malaysia". Also in line 493: "..and (space) Ae. aegypti..."
---

## [Decision Letter · Decision Letter 1]

25 Apr 2023

Dear Dr. Sam,

Thank you very much for submitting your manuscript "Susceptibility of Aedes albopictus, Ae. aegypti and human populations to Ross River virus in Kuala Lumpur, Malaysia" for consideration at PLOS Neglected Tropical Diseases. As with all papers reviewed by the journal, your manuscript was reviewed by members of the editorial board and by several independent reviewers. The reviewers appreciated the attention to an important topic. Based on the reviews, we are likely to accept this manuscript for publication, providing that you modify the manuscript according to the review recommendations. 

Sincerely,

Olaf Horstick, FFPH(UK)

Academic Editor

Elvina Viennet

Section Editor

Reviewer's Responses to Questions

**Key Review Criteria Required for Acceptance?**

**Methods**

-Are the objectives of the study clearly articulated with a clear testable hypothesis stated?

-Is the study design appropriate to address the stated objectives?

-Is the population clearly described and appropriate for the hypothesis being tested?

-Is the sample size sufficient to ensure adequate power to address the hypothesis being tested?

-Were correct statistical analysis used to support conclusions?

-Are there concerns about ethical or regulatory requirements being met?

Reviewer #1: The study has met the acceptance criteria listed in Methods. The aims of the study are clearly stated, and the methods explained here address the stated objectives.

Reviewer #2: Yes

Yes

Yes

I still think the number of samples tested for the seroprevalence part is not enough. However authors indicated this limitation in the discussion part.

Yes

Yes

Reviewer #3: (No Response)

**Results**

-Does the analysis presented match the analysis plan?

-Are the results clearly and completely presented?

-Are the figures (Tables, Images) of sufficient quality for clarity?

Reviewer #1: Results are complete and clearly presented.

Reviewer #2: Results were well presented.

Figure 1: Error bar not visible especially for the Ae. albopictus condition.

 Legend is not complete for the A part.

Reviewer #3: (No Response)

**Conclusions**

-Are the conclusions supported by the data presented?

-Are the limitations of analysis clearly described?

-Do the authors discuss how these data can be helpful to advance our understanding of the topic under study?

-Is public health relevance addressed?

Reviewer #1: Authors mentioned the probable RRV risk in Kuala Lumpur. In abstract as well as in the main text, they have toned up stating Malaysia at high risk of an imported RRV outbreak. However, their own results of sero-prevalence (RRV IgG) in local residents and unavailability of RRV's putative hosts and their vectors other than Ae. albopictus, not really suggest that Malaysia is at a high risk. Better to tone-down the conclusion.

Reviewer #2: Yes

Yes

The discussion part could be summarized.

Reviewer #3: (No Response)

**Editorial and Data Presentation Modifications?**

Reviewer #1: Accept

Reviewer #2: (No Response)

Reviewer #3: (No Response)

**Summary and General Comments**

Reviewer #1: Overall, the work is nicely done, and the paper is executed well. Sero-prevalence of the virus in human population not concerning that clearly suggesting that there is no circulation of RRV in the local setting of the KL. A low tone of RRV risk would better fit for the conclusion.

Reviewer #2: Authors really improved the manuscrit according to the reviewer' comments.

Minor revisions are required for acceptance.

Line 44-46: no link with the previous sentence

Line 52: remove "and is more likely primary vector" because that depend on the vector density also

Line 54: remove "high" before risk

Line 72: modulate the sentence

Method: in Viruses part authors must present the DENV strain used also

Line 522 and 614: mistake in ref 6 and 60 (X))

The discussion must be summarized.

Reviewer #3: (No Response)

PLOS authors have the option to publish the peer review history of their article (what does this mean?). If published, this will include your full peer review and any attached files.

Reviewer #1: No

Reviewer #2: No

Reviewer #3: No

Figure Files:

Data Requirements:

Reproducibility:

References

---

## [Editor Report · Decision Letter 2]

28 May 2023

Dear Dr. Sam,

We are pleased to inform you that your manuscript 'Susceptibility of *Aedes albopictus*, *Ae. aegypti* and human populations to Ross River virus in Kuala Lumpur, Malaysia' has been provisionally accepted for publication in PLOS Neglected Tropical Diseases.

Best regards,

Olaf Horstick, FFPH(UK)

Academic Editor

Elvina Viennet

Section Editor

---

## [Editor Report · Acceptance letter]

7 Jun 2023

Dear Dr. Sam,

We are delighted to inform you that your manuscript, "Susceptibility of *Aedes albopictus*, *Ae. aegypti* and human populations to Ross River virus in Kuala Lumpur, Malaysia," has been formally accepted for publication in PLOS Neglected Tropical Diseases.

Best regards,

Shaden Kamhawi

co-Editor-in-Chief

Paul Brindley

co-Editor-in-Chief
